# Powerless in a Western US Energy Town: Exploring Challenges to Socially Sustainable Rural Development

**Carol J. Ward** [1],*, **Michael R. Cope** [1], **David R. Wilson** [2], **Kayci A. Muirbrook** [1] and **Jared M. Poff** [1]

[1] Department of Sociology, Brigham Young University, Provo, UT 84602, USA; michaelrcope@byu.edu (M.R.C.); kmuirbro@byu.edu (K.A.M.); jpoff2@studentbody.byu.edu (J.M.P.)
[2] Independent Researcher, Springville, UT 84663, USA; hist1700@gmail.com
* Correspondence: carol_ward@byu.edu

**Abstract:** To better understand social sustainability in the context of rapid boom growth and decline, we examine longitudinal social change in the modern boomtown of Colstrip, MT. Using a mixed-methods approach that includes two waves of a community survey—administered in 1996 and 2018, respectively—and focus groups conducted in 2018–2019, we explore shifts in residents' sense of community as well as their perceptions and attitudes about current challenges to their community's future and sustainability. We show that, despite surviving previous boom and bust periods related to changes in the coal industry, this community now faces a new reality that involves the closure of all local power plants. However, both survey responses and residents' narratives indicate a strong sense of community and support for developing strategies that address challenges to the town's future. This exploratory case study helps to extend the literature by contributing to a greater understanding of the experiences of contemporary workers who individually migrated to a small, rural energy town but who now, as a community, face an uncertain future, and by illuminating the role of sense of community in both social and environmental sustainability efforts.

**Keywords:** community; social sustainability; social change

## 1. Introduction

In 1996, researchers revisited the modern boomtown of Colstrip, MT, to assess the long-term effects of the construction and operation of four coal-fired power plants. Now, twenty years later, Colstrip faces a new disruption: closure of the first two power plants. As this rural community confronts the challenges of losing their central source of employment and revenue, this study examines community residents' perceptions of these challenges, their community, interest in staying or leaving, and plans for sustainable community life. To that end, the purposes of this study are threefold: (1) to explore the longitudinal impacts of development on a modern rural energy community, (2) to investigate residents' narratives related to community and sustainability, especially residents' satisfaction with their community and their strategies for development, and (3) to explore residents' perceptions and attitudes about current challenges to their community's future and sustainability. We attend to these goals by employing a mixed methods research design that draws upon key elements of community development, sustainability, socio-spatial, and boomtown research; this study yields insights into how long-time energy communities face new economic realities and challenges. Taken together, this exploratory case study helps to extend the literature on social sustainability by contributing to a greater understanding of the experiences of contemporary workers who once migrated to a small, rural energy town and who now face an uncertain future.

## 2. Background

"Sustainability is no longer about doing less harm. It's about doing more good" [1]. Since the publication of groundbreaking research such as *The Limits to Growth* [2], *Our Common Future* [3], and the United Nations' *Agenda 21* [4], sustainability efforts have indeed set about to do "less harm" and have recently begun to increasingly focus on doing "more good." There are various approaches to sustainability; however, many scholars argue that efforts largely fall into three broad overlapping domains: environmental sustainability, economic sustainability, and social sustainability. Although these domains are interrelated and necessary for achieving sustainability, social sustainability has received less attention. Additionally, despite the growing interest in recent years in social sustainability, "a stable definition has never actually been agreed upon" [5] (p. 612). On one extreme, social sustainability is viewed as the ability of a given community—or group of people—to achieve a positive result related to a given sustainability goal. Increasingly, however, social sustainability is being theorized as the end result of a multifaceted approach to understanding social change and adaptation through the lens of the local community. As such, social sustainability can be understood as a process whereby sustainability is increasing or decreasing. To that end, "[t]he sustainability of community is about the ability of society itself, or its manifestation as local community, to sustain and reproduce itself at an acceptable level of functioning" [6] (p. 293). Consequently, models of social sustainability need to consider temporal processes that capture residents' community experience or "the holistic nature of everyday social interaction articulated in a locality, that which is primarily tied to the locality itself" [7] (p. 434).

A well-known model for assessing social change through the lens of the local community can be found in the boomtown literature. Addressing the effects of social disruption, boomtown research has focused on community to understand the consequences of rapid local community change [8] and the influence that such disruption has on residents' psychological well-being [9]. The boom-bust research also has suggested an eventual recovery stage in such communities, with levels of satisfaction and attachment returning to pre-boom levels [10]. Additionally, rural sociologists have found that communities that stabilized after periods of rapid growth often showed increases in local identity and solidarity [11]. Although boomtown research has focused on these types of changes in energy communities, less attention has been given to their longer-term sustainability. However, recent community sustainability research suggests the relevance and need to explore the perspectives of energy communities on sustainable growth and development.

Colstrip, Montana, provides a case in which to explore this phenomenon. Relatively high investment in community-building amenities laid the groundwork for community growth in a company town originally constructed for coal mining and energy workers. Following the boom and bust cycles associated with building four power plants, the town did not return to pre-boom levels of community attachment as a small ranching community, as some of the boomtown literature suggests (e.g., [10]). Instead, prior research [12] indicates that a new equilibrium was reached in which a sense of community developed independent of the young mining town's economically oriented origins. However, as regional dynamics have shifted away from use of the coal-fired energy produced in Colstrip, the question now concerns the sustainability of this community. Schafft et al. [13] address the importance of the larger context on boomtowns: "Boomtown regions are not undifferentiated recipients of rapid development, but are characterized by a variety of distinct factors," including, among other things, social infrastructure. Certainly, Colstrip offers evidence of this: what began as a company town with the formal Gesellschaft underpinnings has become a Gemeinschaft-like community.

Furthermore, an important feature of the sense of community is its relation to regional—not just local—dynamics, which may threaten the community's future. A socio-spatial approach [14] points to the relevance of interactions with key actors at multiple levels and in multiple contexts. For example, as state, regional, and national political and economic actors' views of, and actions toward, coal-based energy production have become increasingly negative, Colstrip residents' sense of community has intensified, thus strengthening the viability of their community. Nevertheless,

the impact of recent disinvestments by distant company owners no longer interested in buying coal-powered energy suggests the likelihood of a final "bust" for this cohesive community that has successfully weathered previous periods of change. In the face of this uncertain future for the community, residents' narratives express their sense of community and their grief for the possible loss of community. In addition, an intersecting narrative concerns the community's consideration of possible strategies for sustainable local development. Understanding the multi-faceted influences and local reactions to sustaining energy communities such as Colstrip provides valuable insights into community sustainability. While recent research on rural sustainability (e.g., [15]) has focused on narratives of agricultural producers, this paper answers a call by these authors to also study energy producers. Through follow up research on this unique case, we examine the experiences of workers and community residents seeking ways to sustain their community as it addresses new challenges to its stability.

## 2.1. Boomtown Studies: 4 Stages (Pre-boom, Boom, Bust, Recovery)

In their overview of boomtown research, Cope et al. [16] describe the important historical context for energy boomtown development:

In 1973 the Organization of Arab Petroleum Exporting Countries (OAPEC) enacted an oil embargo which effectively quadrupled the price of oil by 1974. Seeking alternative sources, the United States began to expand and explore oil fields within its own borders [9]. Such activities triggered rapid growth in many areas, often doubling community populations within a few years [17].

In the early period of boomtown research (1970s and early 1980s), many of the impact studies focused on the social consequences of significant population growth within the mountainous regions of the Western United States energy production communities [18–22]. The finite lifespans of these "boomtowns" included four stages related to the impact-cycle: pre-boom, boom, bust, and recovery [8,10,23–25].

Focusing on the socio-psychological impacts on residents' attitudes toward their communities [26–30], researchers identified four stages of adjustment to potential changes related to rapid development: enthusiasm, uncertainty, panic, and adaptation [20,24,31–33]. This research also suggests considering both place and time, as levels of residents' attachment to, and satisfaction with, their community are likely to vary and evolve differently along these dimensions.

## 2.2. Community, Social Impact Assessment, and Sustainability

In the 1970s, sociologists and others engaged in research on the West's new boomtowns relied on some form of Tönnies' notion of Gemeinschaft and Gesellschaft [34] as they examined the way, in their view, relatively stable rural communities transformed into impersonal job sites. Much of this research took a deeply pessimistic view of energy resource communities. Many writers either implied or argued that the development of energy resources in the West heralded the replacement of small communities with colonies where anonymity, isolation, and alienation would prevail over rural neighborliness and local attachment.

One of these research projects resulted in an oft-cited minor classic in the sociological literature on rapid development in resource communities (i.e., "boomtowns"): Raymond Gold's [22] A Comparative Case Study of the Impacts of Coal Development on the Way of Life of People in the Coal Areas of Eastern Montana and Northeastern Wyoming. Colstrip achieved a level of notoriety in academic circles because of it. Gold employed qualitative techniques to study Colstrip, Forsyth (an old railroad and agricultural center, and the county seat of Rosebud County where Colstrip is located), and Gillette, Wyoming. After numerous interviews with local residents in the early 1970s, Gold concluded that:

Gillette lacks the sense of community which until recently had always been good in Colstrip and Forsyth. Now, however, the latter's sense of community is definitely breaking down, especially in Colstrip where the proportion of newcomers to established residents is greatest (136).

What Gold observed and documented eventually would be referred to as "social disruption", which became a leading hypothesis and theme of boomtown research throughout the 1970s. Gilmore and Duff [21] echoed many of Gold's conclusions in their study of Rock Springs, Wyoming (cf., [35,36]). These studies purported to document how the conditions labeled "Gillette Syndrome" [37], with its domestic tumult and social anomie, led to the decline of a sense of community in Gillette, Colstrip, and other energy boomtowns.

In 1979, another body of research emerged from the heart of the energy boom region. University of Wyoming sociologist James G. Thompson, alarmed at the "catchy" yet imprecise and largely unconfirmed substance of Kohrs' Gillette Syndrome, tested three of Kohrs' variables in Wyoming energy boomtowns. Two of his three variables, the predicted social pathologies, were not supported. Consequently, Thompson argued that social scientists' use of such terms as Gillette Syndrome was "crowded with emotional imagery" and thus was not useful in developing appropriate public policy responses to the problems these rural communities faced [19].

Wilkinson and his colleagues [38] then suggested that previous community research in Western boomtowns was not only flawed methodologically, but atheoretical. As researchers became increasingly aware of the complexities of influences operating in energy resource communities, they moved away from the old social disruption hypothesis and pursued more complex perspectives [8]. For example, Freudenburg [29] (p. 56) emphasized the development of groups within the community. Other sociologists found stable neighboring patterns in rapidly changing communities [26] (p. 179), and Greider, Krannich, and Berry [11] asserted that a community that has stabilized after its rapid growth may show an increase in local identity and solidarity and the potential for sustainability. In a recent review of boomtown research, Schaff et al. [13] (p. 297) identified three themes: community-level social disorganization associated with rapid local change, particularly strains on local institutions; residents' psychological strain and adjustment to rapid local change; and longitudinal economic effects over the boom–bust–recovery stages of development. These last two themes, in particular, relate to social science literature emphasizing the importance of local communities for individual and collective wellbeing (e.g., see review by Larson et al. [39]).

The research for this paper is related to both the socio-psychological impacts and the longitudinal assessment of the boom–bust–recovery stages. However, this study is also informed by new aspects of boomtown research, specifically a focus on the effects of regional development—economic, demographic, and cultural—on local boomtown communities [13], as well as a socio-spatial approach [14], which addresses the political, economic, and social processes associated with investment and disinvestment in local communities. Finally, this study draws on concepts from the entrepreneurial social infrastructure (ESI) approach to community development [40–42] utilized by Wilson-Forsberg [43] to examine the diverse strategies of four rural Atlantic Canada communities to adapt to change. Specifically, legitimacy of alternatives, resource mobilization, and diverse networks were relevant for understanding the diversity in approaches and outcomes for these communities.

Recent literature suggests several additional theoretical considerations for understanding the continuing impacts of development and the possibilities for sustainability of boomtown communities such as Colstrip. Specifically, in his review of the sustainability literature, Wheeler's [44] (p. 347) discussion of economic development as an action area suggests that sustainable economic development in both small and large places may involve nurturing businesses that "use local resources, have cleaner production practices, pay decent wages, and contribute to the community through civic involvement." Thus, local community initiatives may support sustainable development through expansion of local businesses, innovation incubators, local training, and equity policies.

Other research also suggests the relevance of community as the unit of analysis for understanding sustainability. First, local community context matters in terms of the array of social, physical, economic, and other resources that inform and shape local views and development efforts, and, second, the effects of local actions can be seen more easily at this level [45]. Important to this discussion are the concepts of "epistemic distance" and "epistemic nearness" for analysis of local community views of actions and

their impacts [15,45]. Specifically, these authors show that a localized or community-level focus on sustainable actions often fails to address the possible consequences of local production strategies that may be damaging to more distant regional or global ecosystems. Thus, local narratives advocating a balance between environmental stewardship and profitability may focus on the impacts that are most immediately beneficial at the local level while the more distant (or complex) impacts remain invisible. Conversely, distant decision-makers may not consider the impacts of their actions on the sustainability of energy-producing communities like Colstrip.

In discussing changes in modern communities, Cope et al. [16] assert that although interactions with people in more remote places figure more prominently in modern life, for example, through production and consumption practices, the effects of these interactions are less visible. Thus, awareness of and responsibility for actions affecting extra-local individuals and communities have largely been removed from everyday life. The concept of moral proximity, which refers to social ties that promote social responsibility for the consequences of community-level actions, is useful in addressing in what ways and conditions social ties matter. The implication is that local actions to address strategies that support sustainable development and community well-being must also consider the extra-local—i.e., relationships to and impacts on distant actors and stakeholders. Schafft et al. [13] remind us, however, that while local communities are the sites of significant impacts of larger social and economic processes, local capacities to affect the policies and actions of interest groups and corporate and state agencies typically are quite limited.

Nevertheless, community is important to consider "because it provides the context for human thought, relationship, and action" [46] (p. 14). The narratives of local actors about community as well as approaches to sustainable development shape decision-making and actions that affect both local and extra-local communities and institutions [45,46]. Although the research by Kessler et al. [15] mentioned above examines the discourse among local agricultural producers regarding sustainable practices, it also makes a call to explore discourses on sustainability in relation to other types of producers, such as in the energy industry. Thus, in this paper we explore narratives about sustainable development within the context of a community facing the uncertain future of its main industry, coal-fired energy production. We also discuss how local narratives about community intersect with narratives about strategies for sustainable development.

### 2.3. History of Colstrip

Colstrip was founded in 1924 to supply the Northern Pacific Railroad. By 1958, railroads had converted from coal to diesel locomotives, Colstrip's mine closed, and the railroad sold the property to the Montana Power Company [47].

The construction of four power plants between 1971 and 1984 was a response to the 1970s energy crisis, new EPA regulations that favored the use of low-sulfur coal for energy production, and Puget Sound Energy's need to expand its power supplies to include the use of coal in the Powder River Basin [48]. This resulted in a joint venture with Montana Power Company to build the Colstrip Generating Station (units 1 and 2) and later add units 3 and 4 [48]. Two major population booms resulted from this investment in Colstrip, as well as extraordinary, bordering on sensational, environmental and social controversies. When the energy market leveled off after 1984 and deregulation fever swept the nation in the 1990s, the Montana Power Company divested itself of its energy holdings and sold its properties in Colstrip. In 1998, with the increased role of out-of-state companies operating its power plants and mines, Colstrip residents incorporated what formerly was a "place" into a legally-recognized town.

Colstrip certainly was a boomtown, first in the early 1970s when the first two power plants were built, and again in the 1980s when the second two plants were completed. Colstrip's population in 1970 was 200, in 1971 it more than tripled to 643, and in 1972 it grew to 737 [49] (p. A-68). At the height of construction in 1974, Colstrip's population reached 3900, but it then nose-dived to 1503 in 1980 before the second boom [50]. Importantly, in that same year, when the legal hurdles to building two new

power plants were overcome, both management and labor streamed into Colstrip, and its population rose to 3258. In 1990, the US Census identified 3035 residents [51], but by 2000, the population had declined to 2346 [52].

The Montana Power Company always intended to avoid the image of Rock Springs, Wyoming, where the construction of the Jim Bridger Power Plant in 1970 had turned the former agricultural center and railroad town into a sprawl of trailers and even tents. K. Ross Toole wrote, "It is a town probably gone beyond restoration" [53] (p. 108). Colstrip, unlike Rock Springs, would be planned. Like previous planned company towns, the announced principles of development included several openly self-serving premises: " . . . pleasant living conditions should result in higher employee morale, which should be reflected in job performance"; the public relations problems associated with land reclamation could be addressed through the demonstration of its effectiveness in Colstrip; and it was said that "if Colstrip becomes a model community, the 'company town' image can be a public relations asset rather than a debit" [54] (p. 1–3). Armed with this public relations asset, Montana Power executive Paul Schmechel presented the plans for Colstrip to regional civic groups to reassure them that the new Colstrip would not reproduce the raucous mining camps of the Old West. Schmechel said that the company would retain ownership of the townsite for the time being, " . . . so that we could avoid the tar paper shacks and honky-tonk atmosphere we see so often where construction causes a community to swell overnight" [55] (p. 2).

Noting initial estimates that the power plants would have a life of approximately three decades, Colstrip's landscape contractor posed the hypothetical question of how one plans a community that is anticipated to have an extremely limited life span. The answer he gave: "You don't." Colstrip was designed as a permanent part of the eastern Montana landscape, whether to service Montana Power's energy production or something else. The company argued that effective planning for a pleasant community that serves an energy resource project can, and has, returned benefits to its residents and the corporate sponsor alike [56]. The Montana Power Company invested in the town, including award-winning community planning [54]. In 1978, the National Association of Home Builders and Better Homes and Gardens Magazine awarded the Western Energy Company, the Montana Power subsidiary that managed Colstrip, their Sensible Growth, Design, and Planning Award. Earlier, the Denver Federal Executive Board had conferred on Western Energy its own award for "outstanding achievements" in socioeconomic improvement [57] (p. 1).

During both booms, Colstrip did experience some classic symptoms associated with the social disruption hypothesis. From 1970 to 1975, enrollment in Colstrip schools jumped from 187 to 683 [58]. Classes met wherever room could be found. Schools ran on a double shift, and Colstrip's commercial mall housed classes too. Basic housing needs dominated Colstrip's two booms [12].

Yet processes also were at work in Colstrip that the social disruption hypothesis does not address. A core of permanent residents—not construction workers—began to establish relationships that seemingly run counter to the social disruption hypothesis. For example, The Colstrip Coal and Cattle Country Players, a dinner theater, opened in 1980. The Bechtel Wives, spouses of engineers who had known each other during their husbands' projects from Alaska to Saudi Arabia, formed a kind of welcome wagon to try to offset the isolation of women who followed their husbands to remote worksites.

Soon, new housing and recreation areas were developed, including 31 parks, a 23-mile long walking/running trail, a golf course, and a lake and fishing area. The original schoolhouse was renovated and transformed into a history and art center. A history of the town [12] shows that the community stabilized following the first boom and bust cycle. Despite dropping to just 1503 people in the early 1980s, the population had stabilized at well over 3000 by the end of that decade. In the 1990s, having recovered fully from the first "bust," community members voted to incorporate the town. Now Colstrip Days are celebrated each year, commemorating this company town turned tight-knit community. These efforts represent an important departure from the community planning and design efforts supported by Montana Power; in contrast, they reflect the results of community-based

initiatives—a placemaking process that involved efforts by both local government and residents to create a sustainable quality place "that people want to live, work, play and learn in" [59] (p. 5–7). In fact, these strategic, creative, and tactical placemaking efforts enhance the community's infrastructure, arts, and economic opportunities, which Grabow [60] (p. 5) suggests contribute to livable, sustainable communities. In fact, development of "community-built" projects, such as the history and art museum and other amenities, reflect the emphasis of the placemaking framework on local involvement, which enhances stewardship, community interactions, pride, and attachment [61] (p. 7).

Colstrip now faces a new challenge: the closure of two of its four power plants in 2020 and the other two in 2027 [62]. The current ownership structure complicates the situation, as the majority of owners are located outside of Montana. Owners accelerated the timeline because of declines in the regional market (e.g., Washington and Oregon) for coal-produced energy. However, Montana-based Northwestern Energy has expressed interest in increasing its ownership share of unit 4 [63].

Closures of units 1 and 2 mean the loss of employment and tax revenue for the town as well as a reduction in the school district and community services. Plant owners are negotiating the decommission of units 1 and 2 and any necessary remediation of the land the plant operations occupy. In their analyses of Colstrip's planned transition, Haggerty et al. [64] reported that plans to address the impact of plant closures on Colstrip (residents, city services, infrastructure, etc.) and its transition forward are even less clear than the decommission plans. While the impacts of closing units 1 and 2 are expected to include the loss of some jobs and tax revenues, as well as declines in housing values and city services, these impacts do not necessarily mean the end of Colstrip. While some workers may retire, owners of units 1 and 2 suggest that others will transition to jobs with units 3 and 4 [64]. Thus, layoffs actually may affect only a small proportion of current unit 1 and 2 workers. In contrast, a report from the University of Montana's Bureau of Business and Economic Research [65] on the economic impacts of the early retirement of units 3 and 4 project greater local job and tax revenue losses and other adverse impacts on eastern Montana as well as the entire state and region.

Although local groups, state and national agencies, and environmental groups have expressed different (often opposing) interests in the plant closures and transition forward, the future remains uncertain. However, one thing is clear: the impending plant closures have made more salient the tensions among the interested parties—Colstrip residents, environmental groups, media figures, and outside economic and political actors. An intensified sense of community, expressed as both community attachment and defiance against outside interests, has resulted in advocacy efforts and development of strategies to diversify the economy. Community wellbeing is now a central concern as residents consider how to cope with these challenges to their community.

## 2.4. Summary and Expectations

Recent debates over the importance of sense of community make the social impact assessment controversies of the 1970s and 1980s seem strikingly relevant to today's concerns with sustainable community development, especially persistent boom and bust cycles. More than two decades after boomtown studies captured the attention of many and three decades after the Montana Power Company created a modern boomtown in Colstrip, researchers revisited the town to study the boom and bust phenomenon. Placing the events of the energy crisis era in historical perspective, examining census data, and analyzing a household survey of Colstrip undertaken in 1996, previous research [12,66] addressed whether the so-called "community disruption" of the 1970s led to the community's fragmentation.

New research presented in this paper addresses the impacts of development on this modern rural energy community twenty years later. Of particular interest are narratives related to community and sustainability, especially residents' satisfaction with their community and their strategies for development. The boom–bust–recovery literature suggests that communities may experience a return to pre-boom levels of community attachment and satisfaction as development projects are incorporated into community life and are no longer "new" [10]. However, research on Colstrip suggests that such conclusions may be premature: having survived several smaller boom–bust periods through

the 1990s, Colstrip residents have developed a distinctively strong sense of community as they face new uncertainties about their future. Drawing on key elements of community development, sustainability, socio-spatial, and boomtown research, this study yields insights into how long-time energy communities face new economic realities and challenges.

## 3. Materials and Methods

While many boomtowns populate the Western United States, Colstrip is of particular interest for a number of reasons. First, the timeliness of the research given the community's current state of uncertainty offers a valuable look into local residents' narratives related to community identity, satisfaction, and attachment, as well as transition to sustainable development.

This exploratory mixed methods study draws on both recent qualitative focus group data and survey responses from two time points, 1996 and 2018. The 2018 follow-up study began by capturing the perspectives of those who have invested most in the community. To this end, we conducted three focus groups that included twenty residents who our contact identified as leaders, long-time residents, and local business owners/managers. The first two focus groups were conducted in 2018 and the third in 2019, each lasting approximately three hours. While the first focus group explored a wide range of topics, the second focus group was invited to discuss results of the 2018 survey and provide insights for contextualizing the responses. The third focus group was invited to discuss topics related to both the community's current challenges and prospects for sustainability.

Audio recordings from the focus groups were transcribed using online transcription software, edited for accuracy by the researchers, and then coded for themes. Topics discussed included benefits and drawbacks of life in Colstrip, its history, challenges the community faced, and outsiders' perceptions of Colstrip and prospects for the town's future. The research team's coding yielded key themes, and representative quotes were identified and compiled. Additional perspectives on these themes were obtained from written responses to open-ended survey questions.

This study utilizes a mixed methods approach in which different data sources extend understanding of the community phenomena of interest. To supplement and further contextualize the focus group themes, we present some of the broader community's perspectives using data from the community survey administered in 1996 and the follow-up survey completed between July and October 2018. Survey data for this study come from the Colstrip, Montana, Citizens' Viewpoints survey (CMCV). The CMCV—a cross sectional survey conducted at two points in time—is a one-of-a-kind study that provides a baseline and follow-up data on perceptions of the community, economic and employment concerns, assessments of educational opportunities, and other attributes among residents of Colstrip, MT (see Appendix A for definitions of variable descriptions).

Conducted by community researchers at Brigham Young University (BYU), the CMCV is a traditional paper-based survey that covered a sample of all known households within the community's geographical boundaries. Baseline data were collected in 1996 ($N = 469$), and the survey was administered in person via a door-to-door approach. Residents who consented to participate in the survey were given the option of completing the survey with BYU researchers or having a copy of the questionnaire left at individual residences and picked up the next day. Baseline data represent approximately half of Colstrip households. A second wave of the CMCV was conducted in 2018 ($n = 213$) and represents more than one-fifth of Colstrip households. During this round of data collection, the CMCV again relied on a traditional paper-based survey that covered a sample of all known households within the community's geographical boundaries. However, 2018 data collection efforts differed from the baseline data in that the survey was administered via the mail. In this case, all households were drawn using the U.S. Postal Service's computerized delivery sequence file, which contains all known addresses in Colstrip. In an effort to maximize response rates, 2018 data collection efforts relied on the Dillman multiwave approach [64], consisting of a presurvey contact explaining the study, a second mailing that includes a copy of the questionnaire, and a follow-up reminder mailing. Dillman et al. [67] also recommended that response rates can be improved by

offering multiple ways or modes for respondents to participate. With this recommendation in mind, respondents were given the option to complete the survey via phone or an online version of the survey. Response rates for the CMCV are 56.85% and 22.47%, respectively. We acknowledge a sizable drop in the response rates between 1996 and 2018. This observation is consistent with research from the American Association for Public Opinion Research, which notes that "it was not uncommon for a study conducted twenty years ago to have encountered one refusal for every one or two completed interviews, while today experiencing three or more refusals for every one completed interview is commonplace" [68]. Despite the decline in response rate between the 1996 and 2018 waves of data collection, it should be noted that the 2018 response rate is well above those typically obtained on contemporary surveys by leading research organizations (e.g., Pew Research Center), and is within a range that is typically not a threat to the quality of survey estimates [69–72].

## 4. Results

Following standard practices for focus group studies [73], focus group interviews were designed to obtain detailed information on perspectives of Colstrip residents who have the most experience living in Colstrip and can describe how the community has evolved over the past several decades. Central themes identified in the focus group data analysis began with consideration of the benefits and drawbacks of life in Colstrip. Additional themes related to the community's challenges, misconceptions about Colstrip, ideas about a sustainable path forward, potential impacts of community changes, and community strategies followed. Using survey data that support the purposes of this mixed methods study, statistical analyses of relevant survey results are presented along with focus group themes in the following sections.

### 4.1. Life in Colstrip: Drawbacks and Benefits

Focus group participants discussed their lives in Colstrip, including their initial reactions to moving to a small, highly rural town in the 1980s. Some were skeptical about the decision to relocate there, while others embraced the challenge of working with the mines and power plants. The following comments by two focus group members contrast their initial attitudes with their current feelings about the town:

So, coming to Colstrip, l was like, oh my God, I'm going to nowhere. And I thought I would stay here two years. I taught one year in a rural two-room school so this was a step up . . . I got married here and ended up staying . . . It's a safe town. It's a friendly town.

Another participant commented:

Colstrip basically is our home now. So, there's no place I'd rather live. You know a lot of people ask, "Where are you going to go now that [you] are retired?" . . . This is where our friends are.

Following these comments, focus group participants highlighted the benefits of life in Colstrip, including a sense of belonging, trust, and good quality of life. As one individual explained:

It's a cohesive community . . . When somebody gets sick . . . this town will rally around everybody. You can't believe the amount of money we can raise in one night for somebody that needs care.

Another resident remarked:

Colstrip is so good at, in helping each other as neighbors and forging that bond and that knowledge that when you move here, you're Colstrip. We don't care what your demographic is, we don't care what you, if you're lesbian, gay, any color of the rainbow; it doesn't matter. You're Colstrip, and do right by your neighbors and your neighbors will do right by you. And that's, that's pretty amazing.

Survey responses also showed an increase in mean community satisfaction level from 5.0 in 1996 to 5.75 in 2018, which constitutes an aspect of overall community sentiment. One survey respondent stated, "I've never been more excited about living in Colstrip than I am right now." Others asserted, "It's a great place to raise your kids."

To provide a larger perspective on community sentiment, Table 1 presents the results of an OLS regression of community sentiment over time. Community sentiment is an additive index comprised

of six questions administered both in the 1996 and 2018 surveys, five of which relate to ratings of Colstrip's characteristics on a seven-point Likert scale that ranges from 1, "badly needs improvement", to 7, "exceptional." These questions include rating "access to open places and outdoor activities", "making newcomers feel welcome", "help from others in times of need", "Colstrip as a place to raise a family", and "the friendliness and concern of neighbors". The final question included in the index asked respondents to rate their "overall satisfaction with Colstrip as a place to live" on a seven-point scale on which 1 is dissatisfied and 7 is satisfied. The index ranges from 6 to 42, with a mean of 32.53. The Cronbach's alpha for the index is 0.88, showing that the measure has very good internal reliability. The two survey waves provided longitudinal data that address questions about community sentiment changes over time. Each survey year was dummy-coded to indicate whether the dependent variables differed between the two survey waves. In Table 1, the survey wave from 1996 is used as the reference category for analysis of changes in community sentiment in the data from the 2018 wave.

**Table 1.** OLS regression model predicting change in community sentiment over time.

| | b | | SE | |
|---|---|---|---|---|
| 1996 (reference) | | | | |
| 2018 | 3.260 | *** | 0.575 | |
| | | | | |
| Constant | 31.533 | *** | 0.318 | |
| $R^2$ | 0.047 | | | |
| Adj. $R^2$ | 0.046 | | | |
| *F* | 32.11 | | | |
| *P* | 0.000 | | | |

Notes: $N = 652$. *** $p < 0.001$.

Results of the bivariate OLS regression in Table 1, the equivalent of an analysis of variance, show that there has been a statistically significant increase in community sentiment in Colstrip over time. These data show that current Colstrip residents felt more positive about their community in 2018 compared to community residents in 1996. Linear prediction models calculate the probability of the outcome variable in categories of the explanatory variable. These probabilities were then plotted using estimated means to analyze whether the changes in coefficients between years was statistically significant and not due to variations in sample sizes. Significant differences between time points can be seen through the area of the 95% confidence intervals. The linear prediction models show that community sentiment changed significantly and positively between 1996 and 2018 with a 95% confidence interval. The increase in community sentiment indicates that residents feel more positively about various aspects of their community, both with respect to satisfaction overall and specific community characteristics developed through social networks and attachment.

It is notable that respondents expressed their attachment to, and satisfaction with, the community both with respect to social ties as well as the amenities developed initially by Montana Power Company to establish a sustainable community. For example, one respondent boasted enthusiastically:

We have 31 parks and twenty-three miles of trails and stuff. You go around the lake, you can go anywhere in town on a trail. And you can do it in the dead of winter before they get the streets [cleared].

Colstrip residents also expressed satisfaction with the way in which their community has maintained these amenities:

I mean . . . we've got it going on in Colstrip. We may have an uncertain future, but we have it going on because we are taking the time to take care of our infrastructure now while we have that tax base. So, we will be set up for the next 20 years.

Such expressions emphasize the ways in which deliberate community-building strategies have contributed to the development of a cohesive identity. In this particular case, it may be that community satisfaction has produced, or at least facilitated, a strong community attachment and sense of identity that has helped to outweigh its unique challenges, such as being located in a highly rural area several hours drive from an urban mall, movie theater, and other conveniences.

### 4.2. Challenges for the Community

Closing the mine and the four power plants has begun to have impacts on the community, causing residents to question the sustainability of their community. The mine and oldest plants, units 1 and 2, were closed in 2019, and the other two plants are scheduled to close within the next five years. The effects of these closures included some layoffs and retirements, while most unit 1 and 2 workers were transferred to jobs in units 3 and 4. The focus group participants discussed the ways the loss of employment will affect families and declining tax revenues will impact local institutions, such as the school district and community services. One resident explained, "Well, you shut down [power plants] 1 and 2, your coal trust fund basically drops a significant amount . . . about a third of what [the revenue] you were going on for facilities."

Others expressed fears about the way in which local institutions have handled funds relating to the plant closings:

For a while the school board had not been utilizing the coal board grant and this last year, um, there was a few of us city council members that really went, "Guys, why aren't you doing this?" You're walking away from money that is available to our community that can help cover some of these large expensive items that you've got coming up and we want to make sure those large expensive items get handled now before when units 1 and 2 shutdown.

Although a number of economic challenges, including layoffs, accompanied the earlier bust periods, they were much more sudden than the current situation. As a result, having an actual "expiration date" on Colstrip's main industry is creating a significant degree of unease. One resident voiced:

Fear could be the thing that undermines us right now . . . Good families are leaving, and everybody's like, "Oh should I go . . . should I ride it out?" Because we've got some darn good jobs here, man. We can't recruit new teachers. I know some people are just walking away from their homes.

Such fears stem from the sad realities of other small Montana communities, as expressed by one respondent's personal experience:

You know . . . some towns go, just totally gone. Um, when I was a kid, my dad worked for Livingston before the railroad shut down in Livingston. My family has lived in that area for generations. That town went from a big booming town to almost a ghost town.

Survey data analyses provide additional insights into the qualitative findings on current challenges for the community related to coal-fired energy production. Table 3 shows the results of an OLS bivariate regression of changes in community sentiment over time among workers employed in the mining or power plant industries (those the shutdowns affected most) compared to workers in other sectors. Data on employment in the mining and power plant industries were obtained from the survey question, "How would you describe your work?" Responses that indicated working in the mining or power plant industries were coded 1 and all others were coded 0.

**Table 2.** OLS regression model predicting community sentiment by occupation in mines and power plants.

| | 1996 | | 2018 | | |
|---|---|---|---|---|---|
| | b | SE | b | SE | |
| Other (Reference) Mines/Power Occupation | −0.596 | 0.684 | −2.425 * | 1.130 |  |
| Constant | 31.933 *** | 0.531 | 35.525 *** | 0.620 | |
| N | 453 | | 199 | | |
| $R^2$ | 0.002 | | 0.023 | | |
| Adj. $R^2$ | −0.000 | | 0.018 | | |
| F | 0.84 | | 4.61 | | |
| P | 0.359 | | 0.033 | | |

Notes: * $p < 0.05$, *** $p < 0.001$.

The results show that the community sentiment of respondents working in the mining and power plant industries in 2018 differs significantly from those employed in other sectors. However, the linear prediction model suggests less certainty about differences over time in levels of community sentiment among mine and power plant workers compared to workers in other industries. This can be seen from the overlapping confidence interval for those who are and who are not employed in industries other than the mining and power plant between the two time points. Yet the coefficients vary significantly by year for those employed in the mining and power plant industries, indicating a significant difference between community sentiment for that group compared to others. These findings suggest that while there is general unease about the looming shutdowns, it is impacting the community sentiment of those employed in the affected industries slightly more than persons employed in other sectors. Ancillary regression analyses (available on request) show that these patterns hold, even when controlling for other socio-demographic characteristics.

Community sentiment in Colstrip has strengthened since the population declines following the end of the power plant construction and layoffs in the 1990s. Residents' attitudes about the community are no longer related solely to the employment opportunities the mining and power-generation jobs provided. Residents have developed greater attachment to, and satisfaction with, their community, which could aid in sustaining the community going forward. However, with the recent and upcoming plant and mining shutdowns, further analysis of the boom, bust, and recovery cycle is warranted.

*4.3. Community Identity: Perceptions and Misconceptions of Colstrip*

Environmental sustainability has recently become a focus for community members, especially with attention in the public eye. The focus group participants asserted that environmental groups often misrepresent Colstrip in the media as a backward community of uneducated polluters, and the discussions returned to this theme frequently. "I think people have this impression that we're London in the 1800s," one participant explained. Another respondent claimed, "We do get a lot of media attention . . . most of it's not correct". To illustrate this point, residents spoke of environmentally minded journalists who come to Colstrip early in the morning to take pictures of the power plant smokestacks when the bitter cold makes the steam look like billowing clouds of poisonous gas. The "us vs. them" mentality that has become more prominent in Colstrip during this period of change highlights the strong sense of "us" that the community shares in the first place, much of which is tied to a sense of pride in having consciously built the type of community in which they want to live.

An important theme in the focus groups was related to negative media portrayals of Colstrip that residents feel have left out important information about the mine, the power plants, their employees, and the community itself. One participant explained:

As a community you want the plant and the mine to stay in compliance [with EPA regulations] ... There's ownership [of the operation] by the community ... We live here. Would we go and put "caca" in the air and "caca" in the water and then drink it and breathe it in? No!

Another said, "[Colstrip residents] are fisherman; these are hunters; these are campers. They care. They care about the environment."

While focus group members did acknowledge that pollution of local water aquifers has occurred, they also asserted that remediation or clean-up is the legal responsibility of the mine and plant owners (rather than local citizens and workers). Importantly, this issue was not the focus of their issues with the media, who they believe publish images of the Colstrip area that imply high levels of air pollution by the power plants. As one focus group member explained: "Every picture of Colstrip is an open strip mine and no reclamation, or it's the power plants when it's 20 below and the steam is coming out of the power plants." Thus, the media's lack of attention to the extensive reclamation of mined land and on-going efforts to meet air quality standards continue to fuel local residents' concern about media misrepresentation of Colstrip. However, also missing is acknowledgement by local residents of the more complex impacts of coal-fired energy production on climate change for the region and beyond. Thus, the focus of community narratives relates primarily to local concerns and interests that they may be able to influence.

Colstrip residents also pushed back against the idea that they are opposed to alternative forms of energy: "The kicker here is that a lot of people outside of Colstrip think Colstrip is anti-wind, but we are not anti-wind. But we do want people to understand that we need to walk into this eyes wide open ... That's what we're asking for in reality."

Table **??** displays analyses of survey respondents' attitudes toward environmental issues. These data show that environmental concerns have become more prominent and important in the Colstrip community, highlighting the increasing value placed on environmental sustainability. These concerns and attitudes were evaluated with responses to the question, "Please indicate how much you would approve of clean or environmentally friendly heavy industry and economic development activities for Colstrip." Response options ranged from 1, "strongly disapprove," to 5, "strongly approve." A bivariate OLS regression of respondents' assessments of environmentally friendly industry development in the community and community sentiment between the two time points shows that environmentally-friendly practices were not significant in community sentiment in 1996, but were positively significant in 2018.

**Table 3.** OLS regression model predicting community sentiment by occupation in mines and power plants.

| | 1996 | | 2018 | | |
|---|---|---|---|---|---|
| | **b** | **SE** | **b** | **SE** | |
| Other (Reference) Mines/Power Occupation | −0.596 | 0.684 | −2.425 * | 1.130 | 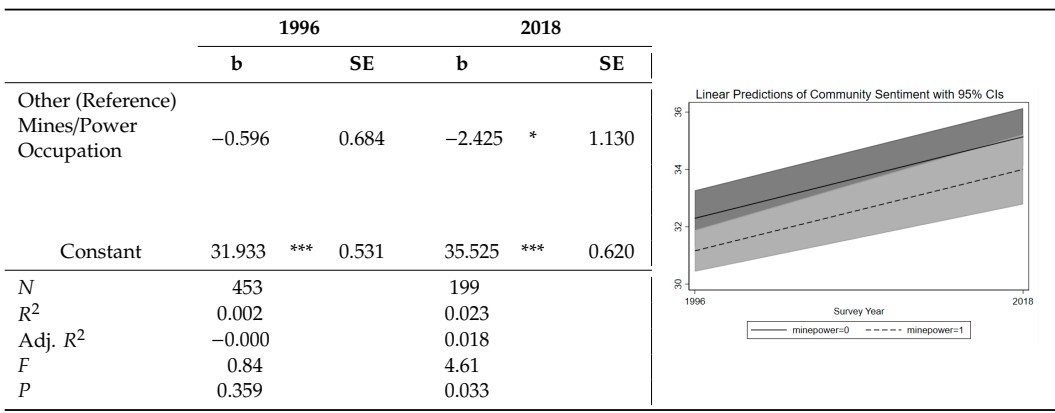 |
| Constant | 31.933 *** | 0.531 | 35.525 *** | 0.620 | |
| *N* | 453 | | 199 | | |
| *R*$^2$ | 0.002 | | 0.023 | | |
| Adj. *R*$^2$ | −0.000 | | 0.018 | | |
| *F* | 0.84 | | 4.61 | | |
| *P* | 0.359 | | 0.033 | | |

Notes: * $p < 0.05$, *** $p < 0.001$.

The results of the analysis show that, in 2018, Colstrip residents who supported more environmentally friendly development also had higher levels of community sentiment, showing that sustainability of the natural environment is important in the community. The more a resident agreed

with this value, the more positive his/her community sentiment. The mean values predicted show that the time period was significant for this attitude: there is a statistically significant difference between the coefficients in 1996 and 2018, as shown by the confidence intervals in the associated graph. Environmentally friendly practices have become more central to community values, further bonding the residents together. The recent negative media portrayal of coal-powered energy in Colstrip has increased awareness and strengthened environmental values. Thus, an important part of the identity that Colstrip residents have constructed is one of stewardship over the land, including going above and beyond requirements that regulators set in order to protect the local area. One distinguishing aspect of the Colstrip operation is that the original owners invested in state-of-the-art technology to remove toxins from the emissions, thanks in large part to pressure from a coalition of local ranchers and the nearby Northern Cheyenne Nation. High standards and strict enforcement hold the mine and power plants accountable to contractual obligations.

The [MT] Department of Environmental Quality oversees what we do ... because the standards they've set for the state of Montana are more stringent than what's being set by the EPA.

An employee of the power plants explained, "If we start up a big diesel engine at the plant, somebody has to be there to take the readings while that's starting up to make sure they've not violated over that six-minute period." The same employee discussed going every six months to get his eyes "re-calibrated" to take visual readings of the plant's emissions.

Of course, Colstrip residents are aware of the complaints and legal actions of area ranchers related to pollutants found in the underground water aquifers. They are also well aware of the land reclamation completed in mined areas. Given the efforts of Colstrip plant and mine workers to comply with environmental policy, they expressed frustrations about their inability to correct what they view as outsiders' damaging misconceptions.

They can lie in the press, and nobody is able to hold them accountable ... If we come back and tell the truth, they just say, "Oh they're just protecting themselves."

Focus group members also believed that positive information about their community is being suppressed in favor of more negative content. One individual complained, "It's like, there is so much science going on in Montana that isn't being talked about."

The difficulties of communicating more accurate information about the community of Colstrip to the broader public have led to the development of groups like Colstrip United, a local grassroots organization that advocates for Colstrip's future and is one of the community's few avenues of public outreach. For example, the organization's Facebook page includes pictures of mined land that has been reclaimed as a way to demonstrate Colstrip's commitment to sustaining the natural environment.

*4.4. A Path Forward?*

Community members also discussed potential avenues through which the community might be sustained. The focus group members who were engineers at the Colstrip power plants estimated that after the mine closes, it will require at least ten to fifteen years to meet the EPA guidelines to restore the environment to its previous state. This includes replacing every plant and stone that was disturbed during the operation, each of which was mapped meticulously before mining could begin, as well as cleaning up the ash ponds, a by-product of the energy production process. Current contracts require that once the power plants have closed, the transmission lines also must be torn out. While these processes may take some time, analyses of survey data (available in Appendix A) show that support for environmentally friendly industry remained relatively high in 2018 compared to 1996 survey responses. Additionally, beliefs that the power plants cause pollution negatively affected community sentiment for some survey respondents, further exemplifying the importance the town places on environmental sustainability. Additionally, stakeholders in Colstrip and others outside the community currently are exploring the possibility of using existing infrastructure to transition to new clean energy sources. For example:

The mine is in negotiations with an outfit to bring in solar—a solar field on mine property. We just don't know how it's gonna work out with connection, with transmission. But they're actually looking at that, and we're not against it.

Another example involves the development of a wind farm north of Colstrip that would use transmission lines that the power plants use currently [74]. Only a small part of the focus group discussion addressed the development of solar energy and other potential solutions. To some extent, this topic was overshadowed by concerns among focus group members about Colstrip's misrepresentation in the media. However, the development of alternative energy sources in Colstrip is being considered by several local, state, and regional actors. For example, Anne Hedges, a member of one of the leading environmental groups that Colstrip residents mentioned, presented an important idea for Colstrip's future in a 2017 Public Broadcasting Service (PBS) documentary [75]:

Here we have a "golden egg" of a transmission system that can move energy to west coast markets. Those markets want clean energy. Somebody will provide them with clean energy. The question is will it be us, or will it be somebody else? Those are the types of conversations we should be having to help the people in Colstrip to move forward, to have employment, and to have certainty.

Others support the idea that the coal-fired energy now produced in Colstrip and transmitted to the Northwest can be replaced by alternative energy sources (e.g., wind farms) developed in Montana [76].

Qualitative responses, as detailed above, show that residents' concerns about negative portrayals of the community may have diverted some attention to finding community-based solutions to the problems associated with the shutdowns. However, focus group comments also indicate support for local organizations or business efforts to explore the development of solar and other energy sources as well as expand local businesses. While focus group participants were generally optimistic about such local energy development, such support was expressed more passively or cautiously, in part, because such large-scale development decisions are outside their control. Instead, the desire for further energy development is expressed in relation to local strategies to lure large energy companies to their community.

For example, one focus group participant expressed excitement about the possibility of attracting large energy companies to Colstrip:

We still have some tremendous opportunities in the energy field . . . the big one I'm excited about is an energy park or an industrial park . . . We seem to have a lot of, you know, a handful, not a lot, but a handful of energy companies that would like to come to Colstrip potentially.

Another option related to outside investment involved bids for the development of wind farms. A number of focus group members expressed interest in this strategy, with one asserting: "They're going to build thousands of those windmills. It'd be nice to build them around here."

However, not all of the discussion of business development centered on outside investment. Another type of development discussed included decisions by local business owners to expand their businesses. Focus group participants gave the example of the local hardware store owner who had been in business for 40 years and recently decided to expand his business to include a lumber yard. One participant's comment represented the views of most participants: "I'm glad he picked up the lumber. Yeah. Oh yeah, that is huge. Doesn't that show some interest and desire to never leave?" Residents expressed special pride in this kind of local investment, with one participant adding, "We've got a lot of tough, resilient people here that are in small businesses . . . I'm just very proud of the activity that this community has shown in the last few years and the investment that is occurring in the last few years, contrary to what everything else is going on."

Both Colstrip insiders and outsiders have identified the development of solar- or wind-generated power as possible answers to their respective concerns about alternatives to the use of coal and Colstrip's future. However, it's unclear whether Colstrip residents are entirely aware of the areas of agreement they share with other groups on these issues.

*4.5. Challenges to Developing a Path Forward*

Even if sources of renewable energy are developed in ways that will benefit the future of Colstrip, residents pointed out challenges related to those potential changes. For example, generational differences in the workforce mean that changing opportunity structures will affect older and younger workers and their families differently. Many community members will retire in the next five to ten years, and it is uncertain what the next generation will do, particularly given the expected mine and power plant closures. While the older generation has experienced a single, long-term career in Colstrip, millennial residents anticipate having several shorter ones. Similarly, while the older generation invested in houses and long-term stability, the younger generation expects to be mobile and less attached to the community in which they work.

As research continues, we will explore their sentiments further. If the longer-term residents of Colstrip acquired their sense of attachment and identity from the degree to which they have invested in the community, it seems unlikely that the younger generation will express this type of attachment and identity, which will present an entirely new set of challenges for the sustainability of the town. However, the town generally maintains a favorable attitude towards their abilities to overcome challenges, despite changing employment and demographics. Ancillary regression analysis (available in Appendix A) shows that residents who have a more positive view of the community's effectiveness in solving problems have higher levels of community sentiment, when controlling for other factors. This speaks to the optimism that many Colstrip residents continue to have, even while facing the upcoming economic changes.

*4.6. Community Strategies*

While Colstrip United represents a formal venue through which residents can resist or attempt to mediate community change, others participate in more informal means, such as patronizing local businesses to show their support. One focus group member who is a long-time resident explained his reason for eating breakfast at the same restaurant all three weekdays on which it is open: "I want to make sure I have a [restaurant] to go to. So, if I don't come, they're not going to have business, and they're going to shut down." There is both practical and symbolic value in such patronage. While it is unlikely that the business of one frequent customer can keep a restaurant open, his action is an expression of investment in the community and helps reinforce a tight-knit community's shared identity. As shown earlier, participants in several focus groups expressed the desire to invest in Colstrip, as illustrated by one local business owner's statement, "Maybe I'm overly optimistic. Colstrip is going to be here and I'm willing to invest in it. It's a great place to live."

## 5. Discussion

Two waves of survey data and data from three focus groups provide insights into how Colstrip residents' satisfaction with and attachment to their community have changed over two decades. Importantly, the socio-economic conditions at these two time points reflects dramatic differences in the market processes that affect the demand for the coal-fired energy produced in Colstrip. While 1996 survey results indicated strong community sentiment and respondents' expectations for continued community prosperity, 2018 results reflected perceptions by local coal-industry workers of the declining demand for coal that was expected to affect their livelihoods and prospects for community wellbeing, despite the results showing strong, positive community sentiment.

Additionally, focus group narratives indicate perceptions and attitudes about current challenges to their community's future and sustainability. Two themes in particular surfaced in Colstrip residents' attitudes and opinions. One is that the degree to which residents are attached to their community has become more salient as threats to that investment materialize. For many respondents, the power plant owners' initial, deliberate investment in the community contributed not only to greater satisfaction, but also to attachment and a cohesive identity, a remarkable development given the relatively short

period of their shared history. Community attachment and identity are expressed in a variety of ways, some of which are defenses against perceived threats from outside, the most salient of which are what local residents believe are outside environmentalists' misrepresentations of energy production in their community. Even the identification of sustainable development solutions, such as local wind or solar power resources, which are strategies of interest to both community members and outsiders, has not received the same level of attention as defense of the community. One reason for this may be that larger scale economic activities are pursued by businesses rather than community residents. However, increasing investment in businesses by local small business owners is a strategy that local residents strongly support. As Wilson-Forsberg [43] (p. 170) discusses, a community may develop reactive solidarity and organize in opposition to outside forces, especially if bridging capital is low or the community does not have sufficient linkages to needed resources.

Of particular importance in the ways in which narratives about community intersect with narratives about sustainability. Narratives generally focused on local community solidarity both in response to perceived outside threats to the sustainability of the community of Colstrip and in relation to the views of possible strategies for sustainable development. For example, in some views, sustainable development will require building on previous community-based projects with new strategic placemaking efforts such as investment in local business growth, as suggested by Grabow [60], Melcher et al. [61], and Wheeler [44]. In this view, local businesses and residents will assume greater responsibility for increasing local economic activity and opportunities that will help to replace jobs lost due to energy plant closures. For others, sustainable development will occur primarily through attracting outside investment in new, more sustainable types of energy development, similar to the model of development of past investment in coal production in Colstrip. Of course, some see sustainable community development as requiring both strategies.

However, because of the current community focus on the threat of impending losses of jobs, services, and homes, current narratives are most attuned to local economic and social impacts of development, regardless of the strategies pursued. As suggested by research on the narratives of other rural producers [15,45], Colstrip residents are more focused on the impacts of local development or production activities on their own community and less so on impacts beyond its borders. For example, although focus group members acknowledged the value of investing in wind and solar energy, the value is largely seen in supporting the local economy and less in terms of the impacts of greener energy sources for the state and region. Another aspect of these narratives is that because local expertise is based in the lengthy experience of workers with extractive industry, knowledge related to pursuing greener energy sources is more limited. For example, when focus group participants expressed enthusiasm and optimism for wind and solar development, one participant with some knowledge of the challenges involved in this type of development strongly advised caution. These findings support those of Brasier et al. [31] that community views of energy development prospects and impacts vary in relation to local level of expertise and community history with development. For Colstrip, the limited local expertise with and financial resources needed to support green energy development results in a sense of powerlessness for pursuing this type of energy production strategy.

A second theme is that the Colstrip community faces an additional challenge in the perceived lower sense of attachment or community identity among young adult residents, which caused some research participants to question the sustainability of strong community attachment. Long-time residents' comments about the younger generation mentioned, for example, their short- vs. long-term financial investments, such as buying ATVs rather than houses. Future research involving younger generation residents will shed more light on these dynamics as the original builders of a tight-knit community attempt to hand over its care to the next generation.

Despite the insights provided by this study, some limitations must be acknowledged. First, as a case study, the findings cannot be generalized to other populations. Nevertheless, the results suggest the relevance of several aspects of the study to examining energy communities. For example, a socio-spatial approach may be useful when considering the ways actors and contexts outside Colstrip

shape both its sense of community and decisions about future sustainable development. Of particular importance are the actions of distant power plant and mine owners who are currently disinvesting in Colstrip's energy production in response to the declining demand for coal-powered energy in regional and global markets. Outside political actors, environmental groups, and the media have contributed to pressures to decommission the mines and power plants.

Community responses to these actions include support for Colstrip as well as actions against the perceived threats to the community's viability. Narratives also suggest more intensive considerations of strategies for pursuing alternative sustainable community development. Although most survey respondents expressed appreciation of their community and concern for its future, there was also some variability in attitudes. For example, some 2018 survey respondents expressed greater interest in environmental issues relative to their sense of community. Focus group participants also supported pursuit of greener energy source development as well as increasing investment in business expansion by local businesses. The strategies expressed in the intersecting narratives about community and sustainable development were largely focused on the local community with much less attention paid to the impacts of these strategies beyond community boundaries. Similar to Kessler et al.'s [15] research findings, the nearness of local impacts is much more visible and compelling to local community members than more distant impacts of local energy production. Another aspect of these narratives is that they do not consider the impacts of development choices for the minority Native American residents, for whom there is a history of prejudice and discrimination. Because the number of Native American survey respondents in 2018 was very small, future focus groups will be organized for Native American residents. A small preliminary focus group conducted in 2018 offers topics for future research to pursue.

Another limitation of this study is related to a fundamental issue for understanding changes in community sentiment: comparison of current residents with those who have left. From the perspective that current residents had a choice and remained in Colstrip, it would be informative to compare current and former residents' perspectives. For example, what differentiates young parents who stayed from those who left? While such comparisons are important, the cross-sectional nature of our data do not allow us to account for the attitudes and opinions of those who relocated out of Colstrip between 1996 and 2018 (although community residents' narratives indicate that most people left as a result of local power plant layoffs and the need to find work elsewhere). To address this limitation, we recommend that future studies on longitudinal social sustainability should consider employing both a panel and a cross-sectional research design.

## 6. Conclusions

On the one hand, this study could be seen as documenting the experiences of contemporary workers who migrated to a small, rural energy town and who face leaving because of the decline of the coal industry. However, such an account misses some unique features that this study makes visible: the decades of placemaking efforts that have resulted in a strong sense of community attachment and ownership among residents. While some residents, out of necessity, have left to find work elsewhere, many of those who remain are working together to develop strategies for sustainability. This strength of community sentiment is illustrated as well by the efforts of former residents to maintain social ties with and support the community's actions, a phenomenon that future research could explore.

Although economic and political actors outside the community will certainly influence decisions about the town's economic future, Colstrip residents are not standing passively on the sidelines. As suggested by the ESI framework discussed by Wilson-Forsberg [43], they are using their substantial resources (human capital and other) to seek ways to diversify their economy and mitigate the economic impacts [77]. Further, Colstrip United is advocating for support at the state and national levels. Although, as Edgel [48] (p. 9) asserted, "Colstrip is an archetype of the coal-dependent community", the town is unique in important ways. Not only has it survived and prospered as an energy town

over a number of decades, its continuing positive community sentiment and community-building experience supports the potential to forge a new, more sustainable future.

**Author Contributions:** The contributions of the authors to development of the research and the manuscript are as follows: conceptualization: C.J.W. and M.R.C.; methodology: C.J.W. and M.R.C.; formal analysis: C.J.W., M.R.C., K.A.M. and J.A.P.; data curation: C.J.W., M.R.C., K.A.M. and J.M.P.; writing—original draft preparation: C.J.W., M.R.C., D.R.W., K.A.M., and J.M.P.; writing—review and editing: C.J.W., M.R.C., D.R.W., K.A.M., and J.M.P.; project administration: C.J.W. and M.R.C.; funding acquisition: C.J.W. and M.R.C. All authors have read and agreed to the published version of the manuscript.

**Funding:** This research was funded in part by two sources internal to Brigham Young University: The Charles Redd Center for Western Studies and a Mentoring Environment Grant. Both of these grants supported student research assistants.

**Acknowledgments:** We would like to acknowledge the contributions of sociologist, J. Lynn England, to the development of the 1996 survey. We would also like to acknowledge assistance from BYU sociology graduate students who contributed to the data collection in the first two focus groups and early drafts of this paper: Jacob Wixom and Meagan Rainock. Additionally, the following graduate students contributed to data analysis of the third focus group: Claudia Soto, Kirstie Weyland, Kirsten Rasmussen, Bret Lyman, Paige Park, Taylor Topham, and Jorden Jackson. Finally, we want to thank the students in the BYU Community Studies Lab for their assistance with the survey datasets.

**Conflicts of Interest:** The authors declare no conflict of interest. The funders had no role in the design of the study; in the collection, analyses, or interpretation of data; in the writing of the manuscript; or in the decision to publish the results.

**Ethics Statement:** Data used in this paper were gathered as part of a study that was reviewed and approved by the Brigham Young University Institutional Review Board. All respondents gave their informed consent prior to their participation in the study.

## Appendix A

**Table A1.** Descriptive Statistics.

| | 1996 (*n* = 469) | | 2018 (*n* = 214) | |
|---|---|---|---|---|
| | **Mean** | **SD** | **Mean** | **SD** |
| Community Sentiment | 31.53 | 6.48 | 34.79 | 7.38 |
| Mine/Power Employment | 67.16% | | 29.44% | |
| Environmentally Friendly Support | 4.07 | 0.93 | 3.92 | 0.90 |
| Community Problem Solving | 2.87 | 1.01 | 3.40 | 1.07 |
| Shutdowns Due to Poor Management | | | 2.09 | 1.13 |
| Shutdowns Due to Lawsuits | | | 4.35 | 1.15 |
| Shutdowns Due to Uninvested Out-of-state Owners | | | 4.13 | 1.13 |
| Units Create too Much Pollution | | | 1.69 | 0.92 |
| Glad Units are Shutting Down | | | 1.40 | 0.96 |
| Management Shows Little Concern | | | 2.55 | 1.20 |
| Male | | | 52.55% | |
| Age | | | 57.13 | 13.39 |
| Length of Residence | | | 0.44 | 0.25 |
| Income | | | $91,483.10 | $38,187.09 |
| White | | | 88.83% | |

Note: Percentages reported for categorical variables.

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
