# Peer review of "Powerless in a Western US Energy Town: Exploring Challenges to Socially Sustainable Rural Development"

_sustainability, doi:10.3390/su12208426_

Round 1

Reviewer 1 Report

Powerless in a Western US Energy Town is an excellent paper! I only have a few minor comments for changes:

1.) Contact Torsten Meyer ([email protected]) regarding additional background lit review on "Boomtown Studies" and "Community, Social Impact Assessment and Sustainability." He has a new book coming out on Boom – Crisis – Heritage. King Coal and the energy revolutions after 1945.

2.) On page 10 (lines 430-441) the author discusses "Current Colstrip residents felt more positive about their community in 2018 than community in 1996" but there was also a significant population drop. The author still needs to account for that people who were less passionate about Colstrip would be more likely to leave the town, which would influence the assessment findings made between 1996 and 2018. A statement regarding this issue needs to be made.

3.) The discussion on "Community Identity: Perceptions and Misconceptions" (pages 13-15, lines 517-598) get into Placemaking Theory and Attachment Theory. The author needs to tie together their findings with the established theories. "Community-Built Art, Construction, Preservation, and Place," edited by Katherine Melcher, Barry Stiefel, Kristin Faurest get into this (https://www.routledge.com/Community-Built-Art-Construction-Preservation-and-Place/Melcher-Stiefel-Faurest/p/book/9781138682580). Other sources could also be consulted on Placemaking Theory and Attachment Theory (many can be found on Google Books)

Author Response

Reviewer 1

  1. The reviewer asked that we contact Thorsten Meyer regarding additional background literature and a forthcoming book. We did so, and received the following reply:

Dear Prof. Ward,

much to our regret the publication of the mentioned conference volume is considerably delayed, it will be published in 2021 only. Unfortunately I could not share any other paper with you.

I kindly ask to apologize this bad news.

Sincerely,

Torsten Meyer

While we were unable to update the literature review based on the reviewer’s comment, we are appreciative for helping make us contact with Dr. Meyer and for making us aware of the forthcoming book. We look forward to the publication of the conference volume and are confident it will help inform our future research.

  1. The reviewer asked that we account for the possibility that people who were less passionate about Colstrip would have been more likely to leave the town which, in turn, would influence analyses of differences between 1996 and 2018. We agree with the reviewer’s assessment. A fundamental issue for understanding changes in community sentiment is that the point of comparison is necessarily with those who have moved. While this is an important comparison, our cross‐sectional data do not permit comparing current residents to those who left; our data to not allow us to comment beyond our sample. Specifically, the cross‐sectional nature of our data limit our ability to account for the attitudes and opinions of those who have relocated out of Colstrip between 1996 and 2018 (although community residents’ narratives indicate that most people left as a result of local job layoffs and the need to find jobs elsewhere). From the perspective that current residents had a choice and made a deliberate decision to remain in Colstrip, as the reviewer notes, it may be more informative to compare current and former residents. For example, what differentiates young parents who stay from those who leave? Cross‐sectional data cannot answer this and related questions. To that end, future studies on longitudinal social sustainability should consider employing both a panel and a cross-sectional research design. Because we cannot empirically speak to the reviewer’s concern, we ultimately felt obliged to forgo an in-depth discussion and instead opted to (1) soften language in findings section, and (2) use this as an opportunity to note an additional limitation of the present study in the Discussion section.

  1. The reviewer asked that we make an effort to tie together our findings with established placemaking and attachment theories. We have endeavored to do so. See additions and edits, for example, in The History of Colstrip, Discussion, and Conclusion sections of the revised manuscript.

Reviewer 2 Report

The article concerns a very current issue of the socio-spatial transformation of an energy town into a sustainable town after the collapse of a power plant. The Colstrip case is valuable because of long-term research observations and the ability to compare results.

General comments

The content of the article should be reorganized or improved to make it more scientific.

  1. The title should better reflect the type and issues of the article.
  2. The abstract should briefly describe the objectives, main problem and conclusions.
  3. The introduction should explain the goals and assumptions first. The content of the Introduction should be shortened and organized. Some parts of it could be removed to constitute a proper discussion.
  4. The discussion is more about conclusions.

Detailed comments

3.4. and 3.5. subsections have the same titles.

Author Response

Reviewer 2

  1. The reviewer noted that the “article should be reorganized or improved to make it more scientific.” We appreciate the reviewer’s comment yet have struggled with the best way to address the concern. While there is a relatively agreed upon format for drafting a report that is a purely quantitative research study or one that is a purely qualitative research study, the scientific standards for reporting on mixed methods research remains less clear (see e.g., Hong, Sutcliffe, and Thomas 2020; Leech 2012). As Collins et al. (2007) note, the problems of representation and legitimation associated with qualitative and quantitative research are exacerbated in mixed methods research “because both the qualitative and quantitative components of studies bring to the study their own unique challenges” (268). Generally speaking, we have endeavored to adhere to the Linear-Analytic report structure and presented the scientific merits of our paper through the use of following ordered sections: introduction, literature review, method, results, and discussion. In so doing, however, we have deliberately chosen to use a written voice and style that embraces subtle realism, or the use of first person and acknowledgement of the researcher (see, e.g., Creswell and Plano Clark 2017; Leech 2012; O’Cathain 2009). As such, we believe our presentation does conform to scientific expectations for reporting mixed method research findings. That said, we agree with the reviewer that some of our choices could be clearer. To that end, in conjunction with points 2-5 below, we have made revisions throughout the manuscript.

  1. The reviewer asked that we adjust the title to better reflect the type and issues of the article. We thank the reviewer for the helpful advice and have made the requested change.

  1. The reviewer asked that we redraft the abstract to briefly describe the objectives, main problem and conclusions. We thank the reviewer for the helpful advice and have made the requested change.

  1. The reviewer noted that the introduction should explain the goals and assumptions first and suggested that the content of the Introduction should be shortened and organized differently. We do so now.

  1. The reviewer asked for a clearer separation between the results and discussion sections. While it is common for mixed method researchers to discuss and contextualize results presented within the findings section (see e.g., Hong, Sutcliffe, and Thomas 2020; Leech 2012), to address the reviewer’s comment, we have made modifications throughout the last half of the revised manuscript.

  1. The reviewer observed that subsections 3.4. and 3.5. have the same titles. This is an oversight on our part. We thank the reviewer for pointing this out; we have renumbered these subsections and corrected the error.

References:

Creswell, John W., and Vicki L. Plano Clark. Designing and conducting mixed methods research. Sage publications, (2017).

Hong, Quan Nha, Rebecca Rees, Katy Sutcliffe, and James Thomas. "Variations of mixed methods reviews approaches: A case study." Research Synthesis Methods (2020).

Leech, Nancy L. "Writing mixed research reports." American Behavioral Scientist 56, no. 6 (2012): 866-881.

O'Cathain, Alicia. "Reporting mixed methods projects." Mixed methods research for nursing and the health sciences (2009): 135-158.

Reviewer 3 Report

Overall, this is an excellent article. It is a clean and convincing study on a moderately well studied topic. The discussion of community identity and sustainability as a comparative during the two time periods is compelling. It might be useful to briefly summarize some of the socio-political differences in the context between the two study periods. This could be a managed in the Discussion section. 

Author Response

Reviewer 3

The reviewer noted that it might be useful to briefly summarize some of the socio-political differences in the context between the two study periods. We thank the reviewer for the helpful insight. With the reviewer’s comment in mind, we have made revisions in for example, in The History of Colstrip, Discussion, and Conclusion sections of the revised manuscript.